# Balancing information exposure in social networks

**Kiran Garimella**
Aalto University & HIIT
Helsinki, Finland
`kiran.garimella@aalto.fi`

**Aristides Gionis**
Aalto University & HIIT
Helsinki, Finland
`aristides.gionis@aalto.fi`

**Nikos Parotsidis**
University of Rome Tor Vergata
Rome, Italy
`nikos.parotsidis@uniroma2.it`

**Nikolaj Tatti**
Aalto University & HIIT
Helsinki, Finland
`nikolaj.tatti@aalto.fi`

## Abstract

Social media has brought a revolution on how people are consuming news. Beyond the undoubtedly large number of advantages brought by social-media platforms, a point of criticism has been the creation of *echo chambers* and *filter bubbles*, caused by *social homophily* and *algorithmic personalization*.

In this paper we address the problem of *balancing the information exposure* in a social network. We assume that two opposing campaigns (or viewpoints) are present in the network, and that network nodes have different preferences towards these campaigns. Our goal is to find two sets of nodes to employ in the respective campaigns, so that the overall information exposure for the two campaigns is *balanced*. We formally define the problem, characterize its hardness, develop approximation algorithms, and present experimental evaluation results.

Our model is inspired by the literature on *influence maximization*, but there are significant differences from the standard model. First, *balance* of information exposure is modeled by a *symmetric difference* function, which is neither monotone nor submodular, and thus, not amenable to existing approaches. Second, while previous papers consider a setting with selfish agents and provide bounds on *best-response* strategies (i.e., move of the last player), we consider a setting with a centralized agent and provide bounds for a global objective function.

## 1 Introduction

Social-media platforms have revolutionized many aspects of human culture, among others, the way people are exposed to information. A recent survey estimates that 62% of adults in the US get their news on social media [15]. Despite providing many desirable features, such as, searching, personalization, and recommendations, one point of criticism is that social media amplify *echo chambers* and *filter bubbles*: users get less exposure to conflicting viewpoints and are isolated in their own informational bubble. This phenomenon is contributed to social homophily and algorithmic personalization, and is more acute for controversial topics [9, 12, 14].

In this paper we address the problem of reducing the filter-bubble effect by balancing information exposure among users. We consider social-media discussions around a topic that are characterized by two or more *conflicting viewpoints*. Let us refer to these viewpoints as *campaigns*. Our approach follows the popular paradigm of influence propagation [18]: we want to select a small number of seed users for each campaign so as to maximize the number of users who are *exposed to both campaigns*. In contrast to existing work on competitive viral marketing, we do not consider the

problem of finding an optimal *selfish strategy* for each campaign separately. Instead we consider a *centralized agent* responsible for balancing information exposure for the two campaigns Consider the following motivating examples.

**Example 1:** Social-media companies have been called to act as arbiters so as to prevent ideological isolation and polarization in the society. The motivation for companies to assume this role could be for improving their public image or due to legislation.[1] Consider a controversial topic being discussed in social-media platform $X$, which has led to polarization and filter bubbles. As part of a new filter-bubble bursting service, platform $X$ would like to disseminate two high-quality and thought-provoking dueling op-eds, articles, one for each side, which present the arguments of the other side in a fair manner. Assume that $X$ is interested in following a viral-marketing approach. Which users should $X$ target, for each of the two articles, so that people in the network are informed in the most balanced way?

**Example 2:** Government organization $Y$ is initiating a program to help assimilate foreigners who have newly arrived in the country. Part of the initiative focuses on bringing the communities of foreigners and locals closer in social media. Organization $Y$ is interested in identifying individuals who can help spreading news of one community into the other.

From the technical standpoint, we consider the following problem setting: We assume that information is propagated in the network according to the *independent-cascade model* [18]. We assume that there are two opposing campaigns, and for each one there is a set of initial seed nodes, $I_1$ and $I_2$, which are not necessarily distinct. Furthermore, we assume that the users in the network are exposed to information about campaign $i$ via diffusion from the set of seed nodes $I_i$. The diffusion in the network occurs according to some information-propagation model.

The objective is to recruit two additional sets of seed nodes, $S_1$ and $S_2$, for the two campaigns, with $|S_1| + |S_2| \leq k$, for a given budget $k$, so as to maximize the expected number of balanced users, i.e., the users who are exposed to information from both campaigns (or from none).

We show that the problem of balancing the information exposure is **NP**-hard. We develop different approximation algorithms for the different settings we consider, as well as heuristic variants of the proposed algorithm. We experimentally evaluate our methods, on several real-world datasets.

Although our approach is inspired by the large body of work on information propagation, and resembles previous problem formulations for competitive viral marketing, there are significant differences. In particular:

- This is the first paper to address the problem of *balancing information exposure* and *breaking filter bubbles*, using the information-propagation methodology.

- The objective function that best suits our problem setting is related to the *size of the symmetric difference* of users exposed to the two campaigns. This is in contrast to previous settings that consider functions related to the *size of the coverage* of the campaigns.

- As a technical consequence of the previous point, our objective function is neither *monotone* nor *submodular* making our problem more challenging. Yet we are able to analyze the problem structure and provide algorithms with approximation guarantees.

- While most previous papers consider selfish agents, and provide bounds on *best-response* strategies (i.e., move of the last player), we consider a centralized setting and provide bounds for a global objective function.

Omitted proofs, figures, and tables are provided as supplementary material. Moreover, our datasets and implementations are publicly available.[2]

## 2 Related Work

**Detecting and breaking filter bubbles.** Several studies have observed that users in online social networks prefer to associate with like-minded individuals and consume agreeable content. This phenomenon leads to *filter bubbles*, *echo chambers* [25], and to online polarization [1, 9, 12, 22].

Once these filter bubbles are detected, the next step is to try to overcome them. One way to achieve this is by making recommendations to individuals of opposing viewpoints. This idea has been explored, in different ways, by a number of studies in the literature [13, 19]. However, previous studies address the problem of breaking filter bubbles by the means of *content recommendation*. To the best of our knowledge, this is the first paper that considers an *information diffusion* approach.

**Information diffusion.** Following a large body of work, we model diffusion using the *independent-cascade model* [18]. In the basic model a single item propagates in the network. An extension is when multiple items propagate simultaneously. All works that study optimization problems in the case of multiple items, consider that items *compete* for being adopted by users. In other words, every user adopts at most one of the existing items and participates in at most one cascade.

Myers and Leskovec [23] argue that spreading processes may either cooperate or compete. Competing contagions decrease each other's probability of diffusion, while cooperating ones help each other in being adopted. They propose a model that quantifies how different spreading cascades interact with each other. Carnes et al. [7] propose two models for competitive diffusion. Subsequently, several other models have been proposed [4, 10, 11, 17, 21, 27, 28].

Most of the work on *competitive information diffusion* consider the problem of selecting the best $k$ seeds for one campaign, for a given objective, in the presence of competing campaigns [3, 6]. Bharathi et al. [3] show that, if all campaigns but one have fixed sets of seeds, the problem for selecting the seeds for the last player is submodular, and thus, obtain an approximation algorithm for the strategy of the last player. Game theoretic aspects of competitive cascades in social networks, including the investigation of conditions for the existence of Nash equilibrium, have also been studied [2, 16, 26].

The work that is most related to ours, in the sense of considering a *centralized authority*, is the one by Borodin et al. [5]. They study the problem where multiple campaigns wish to maximize their influence by selecting a set of seeds with bounded cardinality. They propose a centralized mechanism to allocate sets of seeds (possibly overlapping) to the campaigns so as to maximize the social welfare, defined as the sum of the individual's selfish objective functions. One can choose any objective functions as long as it is submodular and non-decreasing. Under this assumption they provide strategyproof (truthful) algorithms that offer guarantees on the social welfare. Their framework applies for several competitive influence models. In our case, the number of balanced users is not submodular, and so we do not have any approximation guarantees. Nevertheless, we can use this framework as a heuristic baseline, which we do in the experimental section.

## 3    Problem Definition

**Preliminaries:** We start with a directed graph $G = (V, E, p_1, p_2)$ representing a social network. We assume that there are two distinct campaigns that propagate through the network. Each edge $e = (u, v) \in E$ is assigned two probabilities, $p_1(e)$ and $p_2(e)$, representing the probability that a post from vertex $u$ will propagate (e.g., it will be reposted) to vertex $v$ in the respective campaigns.

**Cascade model:** We assume that information on the two campaigns propagates in the network following the independent-cascade model [18]. For instance, consider the first campaign (the process for the second campaign is analogous): we assume that there exists a set of seeds $I_1$ from which the process begins. Propagation proceeds in rounds. At each round, there exists a set of active vertices $A_1$ (initially, $A_1 = I_1$), where each vertex $u \in A_1$ attempts to activate each vertex $v \notin A_1$, such that $(u, v) \in E$, with probability $p_1(u, v)$. If the propagation attempt from a vertex $u$ to a vertex $v$ is successful, we say that $v$ propagates the first campaign. At the end of each round, $A_1$ is set to be the set of vertices that propagated the campaign during the current round.

Given a seed set $S$, we write $r_1(S)$ and $r_2(S)$ for the vertices that are reached from $S$ using the aforementioned cascade process, for the respective campaign. Note that since this process is random, both $r_1(S)$ and $r_2(S)$ are random variables. Computing the expected number of active vertices is a **#P**-hard problem [8], however, we can approximate it within an arbitrary small factor $\epsilon$, with high probability, via Monte-Carlo simulations. Due to this obstacle, all approximation algorithms that evaluate an objective function over diffusion processes reduce their approximation by an additive $\epsilon$. Throughout this work we avoid repeating this fact for the sake of simplicity of the notation.

**Heterogeneous vs. correlated propagations:** We also need to specify how the propagation on the two campaigns interact with each other. We consider two settings: In the first setting, we assume that the campaign messages propagate independently of each other. Given an edge $e = (u, v)$, the vertex $v$ is activated on the first campaign with probability $p_1(e)$, given that vertex $u$ is activated on the first campaign. Similarly, $v$ is activated on the second campaign with probability $p_2(e)$, given that $u$ is activated on the second campaign. We refer to this setting as *heterogeneous*.[3] In the second setting we assume that $p_1(e) = p_2(e)$, for each edge $e$. We further assume that the *coin flips* for the propagation of the two campaigns are totally correlated. Namely, consider an edge $e = (u, v)$, where $u$ is reached by either or both campaigns. Then with probability $p_1(e)$, *any* campaign that has reached $u$, will also reach $v$. We refer to this second setting as *correlated*.

Note that in both settings, a vertex may be active by *none*, *either*, or *both* campaigns. This is in contrast to most existing work in competitive viral marketing, where it is assumed that a vertex can be activated by *at most one* campaign. The intuition is that in our setting activation means merely passing a message or posting an article, and it does not imply full commitment to the campaign. We also note that the heterogeneous setting is more *realistic* than the correlated, however, we also study the correlated model as it is mathematically simpler.

**Problem definition:** We are now ready to state our problem for *balancing information exposure* (BALANCE). Given a directed graph, initial seed sets for both campaigns and a budget, we ask to find additional seeds that would balance the vertices. More formally:

**Problem 3.1** (BALANCE). *Let $G = (V, E, p_1, p_2)$ be a directed graph, and two sets $I_1$ and $I_2$ of initial seeds of the two campaigns. Assume that we are given a budget $k$. Find two sets $S_1$ and $S_2$, where $|S_1| + |S_2| \leq k$ maximizing*

$$\Phi(S_1, S_2) = \mathrm{E}[|V \setminus (r_1(I_1 \cup S_1) \triangle r_2(I_2 \cup S_2))|].$$

The objective function $\Phi(S_1, S_2)$ is the expected number of vertices that are either reached by both campaigns or remain oblivious to both campaigns. Problem 3.1 is defined for both settings, *heterogeneous* and *correlated*. When we need to make explicit the underlying setting we refer to the respective problems by BALANCE-H and BALANCE-C. When referring to BALANCE-H, we denote the objective by $\Phi_H$. Similarly, when referring to BALANCE-C, we write $\Phi_C$. We drop the indices, when we are referring to both models simultaneously.

**Computational complexity:** As expected, the optimization problem BALANCE turns out to be **NP**-hard for both settings, heterogeneous and correlated. A straightforward way to prove it is by setting $I_2 = V$, so the problems reduce to standard influence maximization. However, we provide a stronger result. Note that instead of maximizing balanced vertices we can equivalently minimize the imbalanced vertices. However, this turns to be a more difficult problem.

**Proposition 1.** *Assume a graph $G = (V, E, p_1, p_2)$ with two sets $I_1$ and $I_2$ and a budget $k$. It is an **NP**-hard problem to decide whether there are sets $S_1$ and $S_2$ such that $|S_1| + |S_2| \leq k$ and $\mathrm{E}[|r_1(I_1 \cup S_1) \triangle r_2(I_2 \cup S_2)|] = 0$.*

This result holds for both models, even when $p_1 = p_2 = 1$. This result implies that the minimization version of the problem is **NP**-hard, and there is no algorithm with multiplicative approximation guarantee. It also implies that BALANCE-H and BALANCE-C are also **NP**-hard. However, we will see later that we can obtain approximation guarantees for these maximization problems.

## 4 Greedy algorithms yielding approximation guarantees

In this section we propose three greedy algorithms. The first algorithm yields an approximation guarantee of $(1 - 1/e)/2$ for both models. The remaining two algorithms yield a guarantee for the correlated model only.

**Decomposing the objective:** Recall that the objective function of the BALANCE problem is $\Phi(S_1, S_2)$. In order to show that this function admits an approximation guarantee, we decompose it into two components. To do that, assume that we are given initial seeds $I_1$ and $I_2$, and let us write

$X = r_1(I_1) \cup r_2(I_2), Y = V \setminus X$. Here $X$ are vertices reached by any initial seed in the two campaigns and $Y$ are the vertices that are not reached at all. Note that $X$ and $Y$ are random variables. Since $X$ and $Y$ partition $V$, we can decompose the score $\Phi(S_1, S_2)$ as

$$\Phi(S_1, S_2) = \Omega(S_1, S_2) + \Psi(S_1, S_2), \quad \text{where}$$
$$\Omega(S_1, S_2) = \mathrm{E}[|X \setminus (r_1(I_1 \cup S_1) \triangle r_2(I_2 \cup S_2))|],$$
$$\Psi(S_1, S_2) = \mathrm{E}[|Y \setminus (r_1(I_1 \cup S_1) \triangle r_2(I_2 \cup S_2))|].$$

We first show that $\Omega(S_1, S_2)$ is monotone and submodular. It is well-known that for maximizing a function that has these two properties under a size constraint, the greedy algorithm computes an $(1 - \frac{1}{e})$ approximate solution [24].

**Lemma 2.** $\Omega(S_1, S_2)$ *is monotone and submodular.*

We are ready to discuss our algorithms.

**Algorithm 0: ignore $\Psi$.** Our first algorithm is very simple: instead of maximizing $\Phi$, we maximize $\Omega$, i.e., we ignore any vertices that are made imbalanced during the process. Since $\Omega$ is submodular and monotone we can use the greedy algorithm. If we then compare the obtained result with the empty solution, we get the promised approximation guarantee. We refer to this algorithm as Cover.

**Proposition 3.** *Let $\langle S_1^*, S_2^* \rangle$ be the optimal solution maximizing $\Phi$. Let $\langle S_1, S_2 \rangle$ be the solution obtained via greedy algorithm maximizing $\Omega$. Then*

$$\max\{\Phi(S_1, S_2), \Phi(\emptyset, \emptyset)\} \geq \frac{1 - 1/e}{2} \Phi(S_1^*, S_2^*).$$

**Algorithm 1: force common seeds.** Ignoring the $\Psi$ term may prove costly as it is possible to introduce a lot of new imbalanced vertices. The idea behind the second algorithm is to force $\Psi = 0$. We do this by either adding the same seeds to both campaigns, or adding a seed that is covered by an opposing campaign. This algorithm has guarantees only in the correlated setting with even budget $k$ but in practice we can use the algorithm also for the heterogeneous setting. We refer to this algorithm as Common and the pseudo-code is given in Algorithm 1.

---

**Algorithm 1:** Common, greedy algorithm that only adds common seeds

---

1   $S_1 \leftarrow S_2 \leftarrow \emptyset$;
2   **while** $|S_1| + |S_2| \leq k$ **do**
3      $c \leftarrow \arg\max_c \Phi(S_1 \cup \{c\}, S_2 \cup \{c\})$;
4      $s_1 \leftarrow \arg\max_{s \in I_1} \Phi(S_1, S_2 \cup \{s\})$;
5      $s_2 \leftarrow \arg\max_{s \in I_2} \Phi(S_1 \cup \{s\}, S_2)$;
6      add the best option among $\langle c, c \rangle, \langle \emptyset, s_1 \rangle, \langle s_2, \emptyset \rangle$ to $\langle S_1, S_2 \rangle$ while respecting the budget.

---

We first show in the following lemma that adding common seeds may halve the score, in the worst case. Then, we use this lemma to prove the approximation guarantee

**Lemma 4.** *Let $\langle S_1, S_2 \rangle$ be a solution to* BALANCE-C*, with an even budget $k$. There exists a solution $\langle S_1', S_2' \rangle$ with $S_1' = S_2'$ such that $\Phi_C(S_1', S_2') \geq \Phi_C(S_1, S_2)/2$.*

It is easy to see that the greedy algorithm satisfies the conditions of the following proposition.

**Proposition 5.** *Assume an iterative algorithm where at each iteration, we add one or two vertices to our solution until our constraints are met. Let $S_1^i$, $S_2^i$ be the sets after the $i$-th iteration, $S_1^0 = S_2^0 = \emptyset$. Let $\eta_i = \Phi_C(S_1^i, S_2^i)$ be the cost after the $i$-th iteration. Assume that $\eta_i \geq \eta_{i-1}$. Assume further that for $i = 1, \ldots, k/2$ it holds that $\eta_i \geq \Phi_C(S_1^{i-1} \cup \{c\}, S_2^{i-1} \cup \{c\})$. Then the algorithm yields $(1 - 1/e)/2$ approximation.*

**Algorithm 2: common seeds as baseline.** Not allowing new imbalanced vertices may prove to be too restrictive. We can relax this condition by allowing new imbalanced vertices as long as the gain is at least as good as adding a common seed. We refer this algorithm as Hedge and the pseudo-code is given in Algorithm 2. The approximation guarantee for this algorithm—in the correlated setting and with even budget—follows immediately from Proposition 5 as it also satisfies the conditions.

**Algorithm 2:** Hedge, greedy algorithm, where each step is as good as adding the best common seed

---

**1** $S_1 \leftarrow S_2 \leftarrow \emptyset$;
**2 while** $|S_1| + |S_2| \leq k$ **do**
**3**    $c \leftarrow \arg\max_c \Phi(S_1 \cup \{c\}, S_2 \cup \{c\})$;
**4**    $s_1 \leftarrow \arg\max_s \Phi(S_1, S_2 \cup \{s\})$;
**5**    $s_2 \leftarrow \arg\max_s \Phi(S_1 \cup \{s\}, S_2)$;
**6**    add the best option among $\langle c, c \rangle$, $\langle \emptyset, s_1 \rangle$, $\langle s_2, \emptyset \rangle$, $\langle s_2, s_1 \rangle$, to $\langle S_1, S_2 \rangle$ while respecting the budget.

---

## 5    Experimental evaluation

In this section, we evaluate the effectiveness of our algorithms on real-world datasets. We focus on ($i$) analyzing the quality of the seeds picked by our algorithms in comparison to other heuristic approaches and baselines; ($ii$) analyzing the efficiency and the scalability of our algorithms; and ($iii$) providing anecdotal examples of the obtained results. Although we setup our experiments in order to mimic social behavior, we note that fully realistic experiments would entail the ability to intervene in the network, select seeds, and observe the resulting cascades. This, however, is well beyond our capacity and the scope of the paper.

In all experiments we set $k$ to range between 5 and 50 with a step of 5. We report averages over 1 000 random simulations of the cascade process.

**Datasets:** To evaluate the effectiveness of our algorithms, we run experiments on real-world data collected from twitter. Let $G = (V, E)$ be the twitter follower graph. A directed edge $(u, v) \in E$ indicates that user $v$ follows $u$; note that the edge direction indicates the "information flow" from a user to their followers. We define a cascade $G_X = (X, E_X)$ as a graph over the set of users $X \subseteq V$ who have retweeted at least one hashtag related to a topic (e.g., US elections). An edge $(u, v) \in E_X \subseteq E$ indicates that $v$ retweeted $u$.

We use datasets from six topics with opposing viewpoints, covering politics (US-elections, Brexit, ObamaCare), policy (Abortion, Fracking), and lifestyle (iPhone, focusing on iPhone vs. Samsung). All datasets are collected by filtering the twitter streaming API (1% random sample of all tweets) for a set of keywords used in previous work [20]. For each dataset, we identify two sides (indicating the two view-points) on the retweet graph, which has been shown to capture best the two opposing sides of a controversy [12]. Details on the statistics of the dataset can be found at the supplementary material.

After building the graphs, we need to estimate the diffusion probabilities for the heterogeneous and correlated models. Note that the estimation of the diffusion probabilities is orthogonal to our contribution in this paper. For the sake of concreteness we have used the approach described below. One could use a different, more advanced, method; our methods are still applicable.

Let $q_1(v)$ and $q_2(v)$ be an *a priori* probability of a user $v$ retweeting sides 1 and 2, respectively. These are measured from the data by looking at how often a user retweets content from users and keywords that are discriminative of each side. For example, for US-elections, the discriminative users and keywords for side Hillary would be @hillaryclinton and #imwither, and for Trump, @realdonaldtrump and #makeamericagreatagain. The probability that user $v$ retweets user $u$ (cascade probability) is then defined as

$$p_i(u, v) = \alpha \, q_i(v) + (1 - \alpha) \left( \frac{R(u, v) + 1}{R(v) + 2} \right), \quad i = 1, 2,$$

where $R(u, v)$ is the number of times $v$ has retweeted $u$, and $R(v)$ is the total number of retweets of user $v$. The cascade probabilities $p_i$ capture the fact that users retweet content if they see it from their friends (term $\frac{R(u,v)+1}{R(v)+2}$) or based on their own biases (term $q_i(v)$). The additive terms in the numerator and denominator provide an additive smoothing by Laplace's rule of succession.

We set the value of $\alpha$ to 0.8 for the heterogeneous setting. For $\alpha = 0$ the edge probabilities become equal for the two campaigns, which is our assumption for the correlated setting.

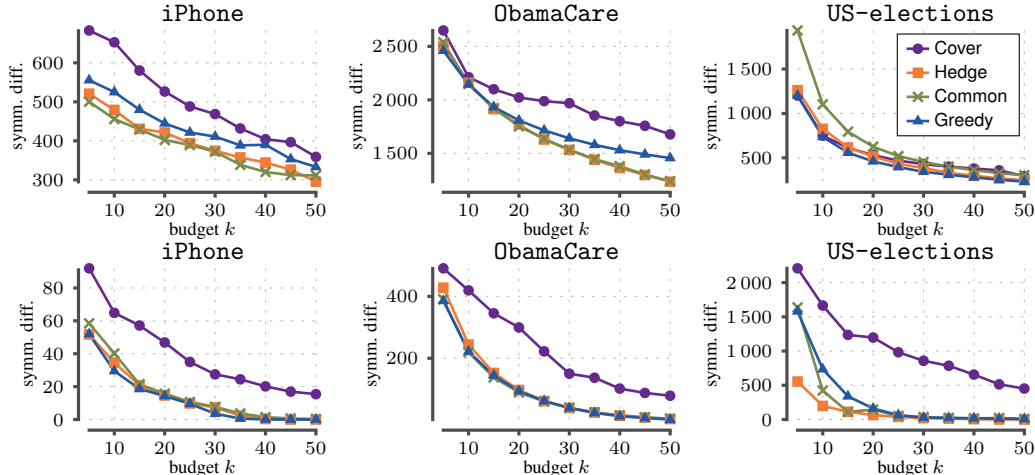

Figure 1: Expected symmetric difference $n - \Phi_C$ as a function of the budget $k$. Top row, heterogeneous model, bottom row: Correlated model. Low values are better.

**Baselines.** We use 5 different baselines. The first baseline, BBLO, is an adaptation of the framework by Borodin et al. [5]. This framework requires an objective function as input, and here we use our objective function $\Phi$. The framework works as follows: The two campaigns are given a budget $k/2$ on the number of seeds that they can select. At each round, we select a vertex $v$ for $S_1$, optimizing $\Phi(S_1 \cup \{v\}, S_2)$, and a vertex $w$ for $S_2$, optimizing $\Phi(S_1, S_2 \cup \{w\})$. We should stress that the theoretical guarantees by [5] do not apply because our objective is not submodular.

The next two heuristics add a set of common seeds to both campaigns. We run a greedy algorithm for campaign $i = 1, 2$ to select the set $S'_i$ with the $\ell \gg k$ vertices $P_i$ that optimizes the function $r_i(S'_i \cup I_i)$. We consider two heuristics: Union selects $S_1$ and $S_2$ to be equal to the $k/2$ first distinct vertices in $S'_1 \cup S'_2$ while Intersection selects $S_1$ and $S_2$ to be equal to $k/2$ first vertices in $S'_1 \cap S'_2$. Here the vertices are ordered based on their discovery time.

Finally, HighDegree selects the vertices with the largest number of followers and assigns them alternately to the two cascades; and Random assigns $k/2$ random seeds to each campaign.

In addition to the baselines, we also consider a simple greedy algorithm Greedy. The difference between Cover and Greedy is that, in each iteration, Cover adds the seed that maximizes $\Omega$, while Greedy adds the seed that maximizes $\Phi$. We can only show an approximation guarantee for Cover but Greedy is a more intuitive approach, and we use it as a heuristic.

**Comparison of the algorithms.** We start by evaluating the quality of the sets of seeds computed by our algorithms, i.e., the number of equally-informed vertices.

*Heterogeneous setting.* We consider first the case of heterogeneous networks. The results for the selected datasets are shown in Figure 1. Full results are shown in the supplementary material. Instead of plotting $\Phi$, we plot the number of the remaining unbalanced vertices, $n - \Phi$, as it makes the results easier to distinguish; i.e., an optimal solution achieves the value 0.

The first observation is that the approximation algorithm Cover performs, in general, worse than the other two heuristics. This is due to the fact that Cover does not optimize directly the objective function. Hedge performs better than Greedy, in general, since it examines additional choices to select. The only deviation from this picture is for the US-elections dataset, where the Greedy outperforms Hedge by a small factor. This may due to the fact that while Hedge has more options, it allocates seeds in batches of two.

*Correlated setting.* Next we consider correlated networks. We experiment with the three approximation algorithms Cover, Common, Hedge, and the heuristic Greedy. The results are shown in Figure 1. Cover performs again the worst since it is the only method that introduces new unbalanced vertices without caring about their cardinality. Its variant, Greedy, performs much better in practice even though it does not provide an approximation guarantee. The algorithms Common, Greedy, and Hedge perform very similar to each other without a clear winner.

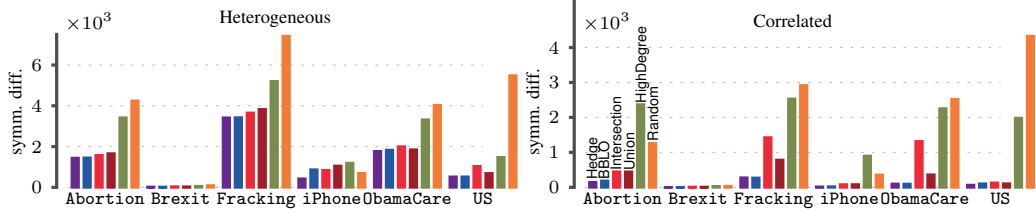

Figure 2: Expected symm. diff. $n - \Phi$ of Hedge and the baselines. $k = 20$. Low values are better.

**Comparison with baselines.** Our next step is to compare against the baselines. For simplicity, we focus on $k = 20$; the overall conclucions hold for other budgets. The results for Hedge versus the five baselines are shown in Figure 2.

From the results we see that BBLO is the best competitor: its scores are the closest to Hedge, and it receives slightly better scores in 3 out of 12 cases. The competitiveness is not surprising because we specifically set the objective function in BBLO to be $\Phi(S_1, S_2)$. The Intersection and Union also perform well but are always worse than Hedge. Random is unpredictable but always worse than Hedge. In the case of heterogeneous networks, Hedge selects seeds that leave less unbalanced vertices, by a factor of two on average, compared to the seeds selected by the HighDegree method. For correlated networks, our method outperforms the two baselines by an order of magnitude. The actual values of this experiment can be found in the supplementary material.

**Running time.** We proceed to evaluate the efficiency and the scalability of our algorithms. We observe that all algorithms have comparable running times and good scalability. More information can be found in the supplementary material.

**Use case with Fracking.** We present a qualitative case-study analysis for the seeds selected by our algorithm. We highlight the Fracking dataset, even though we applied similar analysis to the other datasets as well (the results are given in the supplementary material of the paper). Recall that for each dataset we identify two sides with opposing views, and a set of initial seeds for each side ($I_1$ and $I_2$). We consider the users in the initial seeds $I_1$ (side supporting fracking), and summarize the text of all their Twitter profile descriptions in a word cloud. The result, contains words that are used to emphasize the benefits of fracking (energy, oil, gas, etc.). We then draw a similar word cloud for the users identified by the Hedge algorithm as seed nodes in the sets $S_1$ and $S_2$ ($k = 50$). The result, contains a more balanced set of words, which includes many words used to underline the environmental dangers of fracking. We use word clouds as a qualitative case study to complement our quantitative results and to provide more intuition about our problem statement, rather than an alternative quantitative measure.

## 6 Conclusion

We presented the first study of the problem of balancing information exposure in social networks using techniques from the area of information diffusion. Our approach has several novel aspects. In particular, we formulate our problem by seeking to optimize a *symmetric difference* function, which is neither monotone nor submodular, and thus, not amenable to existing approaches. Additionally, while previous studies consider a setting with selfish agents and provide bounds on best-response strategies (i.e., move of the last player), we consider a centralized setting and provide bounds for a global objective function.

Our work provides several directions for future work. One interesting problem is to improve the approximation guarantee for the problem we define. Second, we would like to extend the problem definition for more than two campaigns and design approximation algorithms for that case. Finally, we believe that it is worth studying the BALANCE problem under complex diffusion models that capture more realistic social behavior in the presence of multiple campaigns. One such extension is to consider propagation probabilities on the edges that are dependent in the past behavior of the nodes with respect to the two campaigns, e.g., one could consider Hawkes processes [28].

**Acknowledgments.** This work has been supported by the Academy of Finland projects "Nestor" (286211) and "Agra" (313927), and the EC H2020 RIA project "SoBigData" (654024).

## Footnotes

[1]For instance, Germany is now fining Facebook for the spread of fake news.

[2]`https://users.ics.aalto.fi/kiran/BalanceExposure/`

[3]Although *independent* is probably a better term than *heterogeneous*, we adopt the latter to avoid any confusion with the independent-cascade model.

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
