[Supplementary Material · supplementary-material.pdf]

# Balancing information exposure in social networks

## Supplementary material

**Kiran Garimella**
Aalto University & HIIT
Helsinki, Finland
kiran.garimella@aalto.fi

**Aristides Gionis**
Aalto University & HIIT
Helsinki, Finland
aristides.gionis@aalto.fi

**Nikos Parotsidis**
University of Rome Tor Vergata
Rome, Italy
nikos.parotsidis@uniroma2.it

**Nikolaj Tatti**
Aalto University & HIIT
Helsinki, Finland
nikolaj.tatti@aalto.fi

## A  Proof of Proposition 1

*Proof.* To prove the hardness we will use SET COVER. Here, we are given a universe $U$ and family of sets $C_1, \ldots, C_\ell$, and we are asked to select $k$ sets covering the universe $U$.

To map this instance to our problem, we first define vertex set $V$ to consist of 3 parts, $V_1$, $V_2$ and $V_3$. The first part corresponds to the universe $U$. The second part consists of $k$ copies of $\ell$ vertices, $i$th vertex in $j$th copy corresponds to $C_i$. The third part consists of $k$ vertices $b_j$. The edges are as follows: a vertex $v$ in the $j$th copy, corresponding to a set $C_i$ is connected to the vertices corresponding to the elements in $C_i$, furthermore $v$ is connected to $b_j$. We set $p_1 = p_2 = 1$. The initial seeds are $I_1 = \emptyset$ and $I_2 = V_1 \cup V_3$. We set the budget to $2k$.

Assume that there is a $k$-cover, $C_{i_1}, \ldots, C_{i_k}$. We set

$$S_1 = S_2 = \left\{ \text{vertex corresponding to } C_{i_j} \text{ in } j\text{th copy} \right\}.$$

It is easy to see that the imbalanced vertices in $I_2$ are exposed to the first campaign. Moreover, $S_1$ and $S_2$ do not introduce new imbalanced vertices. This makes the objective equals to 0.

Assume that there exists a solution $S_1$ and $S_2$ with a zero cost. We claim that $|S_1 \cap (V_1 \cup V_2)| \le k$. To prove this, first note that $S_1 \cap V_2 = S_2 \cap V_2$, as otherwise vertices in $V_2$ are left unbalanced. Let $m = |S_1 \cap V_2|$. Since $V_3$ must be balanced and each vertex in $V_2$ has only one edge to a vertex in $V_3$, there at least $k$ vertices in $|S_1 \cap \{V_2 \cup V_3\}|$, that is, we must have $|S_1 \cap V_3| \ge k - m$. Let us write $d_{ij} = |S_i \cap V_j|$. The budget constraints guarantee that

$$d_{11} + d_{12} + d_{22} + d_{13} \le \sum_{ij} d_{ij} \le 2k,$$

which can be rewritten as

$$d_{11} + d_{12} \le 2k - d_{22} - d_{13} \le 2k - m - (k - m) = k.$$

Construct $C$ as follows: for each $S_1 \cap V_2$, select the set that correponds to the vertex, for each $S_1 \cap V_1$, select any set that contain this vertex (there is always at least one set, otherwise the problem is trivially false). Since $V_1$ must be balanced, $C$ is a $k$-cover of $U$.  $\square$

## B  Proof of Lemma 2

Before providing the proof, as a technicality, note that submodularity is usually defined for functions with one argument. Namely, given a universe of items $U$, we consider functions of the type $f : 2^U \to \mathbb{R}$. However, by taking $U = V \times \{1, 2\}$ we can equivalently write our objectives as functions with one argument, i.e., $\Phi, \Omega, \Psi : 2^U \to \mathbb{R}$.

*Proof.* The objective counts 3 types of vertices: (*i*) vertices covered by both initial seeds, (*ii*) additional vertices covered by $I_1$ and $S_2$, and (*iii*) additional vertices covered by $I_2$ and $S_1$. This allows us to decompose the objective as

$$\Omega(S_1, S_2) = \mathrm{E}[|A| + |B| + |C|], \quad \text{where}$$

$$A = r_1(I_1) \cap r_2(I_2), \quad B = (r_1(I_1) \setminus r_2(I_2)) \cap r_2(S_2), \quad C = (r_2(I_2) \setminus r_1(I_1)) \cap r_1(S_1).$$

Note that $A$ does not depend on $S_1$ and $S_2$. $B$ grows in size as we add more vertices to $S_2$, and $C$ grows in size as we add more vertices to $S_1$. This proves that the objective is monotone.

To prove the submodularity, let us introduce some notation: given a set of edges $F$, we write $r(S; F)$ to be the set of vertices that can be reached from $S$ via $F$. This allows us to define

$$A(F_1, F_2) = r(I_1; F_1) \cap r(I_2; F_2),$$
$$B(F_1, F_2) = (r(I_1; F_1) \setminus r(I_2; F_2)) \cap r(S_2; F_2),$$
$$C(F_1, F_2) = (r(I_2; F_2) \setminus r(I_1; F_1)) \cap r(S_1; F_1).$$

The score $\Omega(S_1, S_2)$ can be rewritten as

$$\sum_{F_1, F_2} p(F_1, F_2)(|A(F_1, F_2)| + |B(F_1, F_2)| + |C(F_1, F_2)|),$$

where $p(F_1, F_2)$ is the probability of $F_1$ being the realization of the edges for the first campaign and $F_2$ being the realization of the edges for the second campaign.

The first term $A(F_1, F_2)$ does not depend on $S_1$ or $S_2$. The second term is submodular as a function of $S_2$ and does not depend of $S_1$. The third term is submodular as a function of $S_1$ and does not depend of $S_2$. Since any linear combination of submodular function weighted by positive coefficients is also submodular, this completes the proof. $\qquad\square$

## C  Proof of Proposition 3

*Proof.* Write $c = 1 - 1/e$. Let $\langle S_1', S_2' \rangle$ be the optimal solution maximizing $\Omega$. Lemma 2 shows that $\Omega(S_1, S_2) \geq c\Omega(S_1', S_2')$.

Note that $\Psi(\emptyset, \emptyset) \geq \Psi(S_1^*, S_2^*)$ as the first term is the average of vertices not affected by the initial seeds. Thus,

$$\begin{aligned}
\Phi(S_1^*, S_2^*) = \Omega(S_1^*, S_2^*) + \Psi(S_1^*, S_2^*) &\leq \Omega(S_1', S_2') + \Psi(S_1^*, S_2^*) \\
&\leq \Omega(S_1', S_2') + \Psi(\emptyset, \emptyset) \leq \Omega(S_1, S_2)/c + \Psi(\emptyset, \emptyset) \\
&\leq \Omega(S_1, S_2)/c + \Psi(\emptyset, \emptyset)/c \\
&\leq (2/c) \max\{\Omega(S_1, S_2), \Psi(\emptyset, \emptyset)\} \\
&\leq (2/c) \max\{\Phi(S_1, S_2), \Phi(\emptyset, \emptyset)\},
\end{aligned}$$

which completes the proof. $\qquad\square$

## D  Proof of Lemma 4

*Proof.* As we are dealing with the correlated setting, we can write $r(S) = r_1(S) = r_2(S)$. Our first step is to decompose $\omega = \Phi_C(S_1, S_2)$ into several components. To do so, we partition the vertices based on their reachability from the initial seeds,

$$\begin{aligned}
A &= r(I_1) \cap r(I_2), & B &= r(I_1) \setminus r(I_2), \\
C &= r(I_2) \setminus r(I_1), & D &= V \setminus (r(I_1) \cup r(I_2)).
\end{aligned}$$

Note that these are all random variables. If $S_1 = S_2 = \emptyset$, then $\Phi_C(S_1, S_2) = \mathrm{E}_C[|A| + |D|]$. More generally, $S_1$ may balance some vertices in $C$, and $S_2$ may balance some vertices in $B$. We may also introduce new imbalanced vertices in $D$. To take this into account we define

$$B' = B \cap r(S_2), \qquad\qquad C' = C \cap r(S_1),$$
$$D' = D \setminus (r(S_1) \triangle r(S_2)).$$

We can express the cost of $\Phi_C(S_1, S_2)$ as

$$\omega = \Phi_C(S_1, S_2) = \mathrm{E}_C[|A| + |B'| + |C'| + |D'|].$$

Split $S_1 \cup S_2$ in two equal-size sets, $T$ and $Q$, and define

$$\omega_1 = \Phi_C(T, T), \quad \omega_2 = \Phi_C(Q, Q).$$

We claim that $\omega \le \omega_1 + \omega_2$. This proves the proposition, since $\omega_1 + \omega_2 \le 2 \max\{\omega_1, \omega_2\}$.

To prove the claim let us first split $T$ and $Q$,

$$T_1 = T \cap S_1, \ T_2 = T \cap S_2, \ Q_1 = Q \cap S_1, \ Q_2 = Q \cap S_2.$$

Our next step is to decompose $\omega_1$ and $\omega_2$, similar to $\omega$. To do that, we define

$$B_1 = B \cap r(T_2), \qquad\qquad B_2 = B \cap r(Q_2),$$
$$C_1 = C \cap r(T_1), \qquad\qquad C_2 = C \cap r(Q_1).$$

Note that, the pair $\langle T, T \rangle$ does not introduce new imbalanced nodes. This leads to

$$\omega_1 = \Phi_C(T, T) = \mathrm{E}_C[|A| + |B_1| + |C_1| + |D|],$$

and similarly,

$$\omega_2 = \Phi_C(Q, Q) = \mathrm{E}_C[|A| + |B_2| + |C_2| + |D|].$$

To prove $\omega \le \omega_1 + \omega_2$, note that $|D'| \le |D|$. In addition,

$$|B'| = |B \cap (r(T_2) \cup r(Q_2))|$$
$$\le |B \cap r(T_2)| + |B \cap r(Q_2)| = |B_1| + |B_2|$$

and

$$|C'| = |C \cap (r(T_1) \cup r(Q_1))|$$
$$\le |C \cap r(T_1)| + |C \cap r(Q_1)| = |C_1| + |C_2|.$$

Combining these inequalities proves the proposition. $\qquad\qquad\qquad\qquad\qquad\square$

## E   Proof of Proposition 5

To prove the proposition, we need the following technical lemma, which is a twist of a standard technique for proving the approximation ratio of the greedy algorithm on submodular functions.

**Lemma 1.** *Assume a universe $U$. Let $f : 2^U \to \mathbb{R}$ be a positive function. Let $T \subseteq U$ be a set with $k$ elements. Let $C_0 \subseteq \cdots \subseteq C_k$ be a sequence of subsets of $U$. Assume that $f(C_i) \ge \max_{t \in T} f(C_{i-1} \cup \{t\})$.*

*Assume further that for each $i = 1, \ldots, k$, we can decompose $f$ as $f = g_i + h_i$ such that*

    *1. $g_i$ is submodular and monotonically increasing function,*

    *2. $h_i(W) = h_i(C_{i-1})$, for any $W \subseteq T \cup C_{i-1}$.*

*Then $f(C_k) \ge (1 - 1/e)f(T)$.*

*Proof.* The assumptions of the propositions imply

$$
\begin{aligned}
f(T) &= g_i(T) + h_i(T) \\
&= g_i(T) + h_i(C_{i-1}) \\
&\le g_i(C_{i-1}) + h_i(C_{i-1}) + \sum_{t \in T} g_i(C_{i-1} \cup \{t\}) - g_i(C_{i-1}) \\
&= f(C_{i-1}) + \sum_{t \in T} h_i(C_{i-1}) + g_i(C_{i-1} \cup \{t\}) - g_i(C_{i-1}) - h_i(C_{i-1}) \\
&= f(C_{i-1}) + \sum_{t \in T} h_i(C_{i-1} \cup \{t\}) + g_i(C_{i-1} \cup \{t\}) - g_i(C_{i-1}) - h_i(C_{i-1}) \\
&= f(C_{i-1}) + \sum_{t \in T} f(C_{i-1} \cup \{t\}) - f(C_{i-1}) \\
&\le f(C_{i-1}) + k(f(C_i) - f(C_{i-1})),
\end{aligned}
$$

where the first inequality is due to the submodularity of $g_i$, and is a standard trick to prove the approximation ratio for the greedy algorithm.

We can rewrite the above inequality as

$$
kf(T) + (1-k)f(T) = f(T) \le f(C_{i-1}) + k(f(C_i) - f(C_{i-1})).
$$

Rearranging the terms leads to

$$
\frac{k-1}{k}(f(C_{i-1}) - f(T)) \le f(C_i) - f(T) \quad .
$$

Applying induction over $i$, yields

$$
f(C_k) - f(T) \ge \left(\frac{k-1}{k}\right)^k (f(C_0) - f(T)) \ge \frac{1}{e}(f(C_0) - f(T)) \ge -f(T)/e,
$$

leading to $f(C_k) \ge (1 - 1/e)f(T)$. $\qquad\square$

We can now prove the main claim. Note that since we are using the correlated model, we have $r_1 = r_2$. For notational simplicity, we will write $r = r_1 = r_2$.

*Proof of Proposition 5.* Let $OPT$ be the cost of the optimal solution. Let $D$ be the solution maximizing $\Phi_C(D, D)$ with $|D| \le k/2$. Lemma 4 guarantees that $OPT/2 \le \Phi_C(D, D)$.

In order to apply Lemma 6, we first define the universe $U$ as

$$
U = \{\langle u, v \rangle \mid u, v \in V\} \cup \{\langle v, \emptyset \rangle \mid v \in V\} \cup \{\langle \emptyset, v \rangle \mid v \in V\}.
$$

The sets are defined as

$$
C_i = \left\{\langle v, \emptyset \rangle \mid v \in S_1^i\right\} \cup \left\{\langle \emptyset, v \rangle \mid v \in S_2^i\right\}.
$$

Given a set $C \subseteq U$, let us define $\pi_1(C) = \{v \mid \langle v, u \rangle \in C, v \ne \emptyset\}$ to be the union of the first entries in $C$. Similarly, define $\pi_2(C) = \{v \mid \langle u, v \rangle \in C, v \ne \emptyset\}$.

We can now define $f$ as $f(C) = \Phi_C(\pi_1(C), \pi_2(C))$. To decompose $f$, let us first write

$$
X_i = r(I_1 \cup \pi_1(C_{i-1})) \cup r(I_2 \cup \pi_2(C_{i-1})) = r(I_1 \cup S_1^{i-1}) \cup r(I_2 \cup S_2^{i-1}), \quad Y_i = V \setminus X_i.
$$

and set

$$
\begin{aligned}
g_i(C) &= \mathrm{E}[|X_i \setminus (r(I_1 \cup \pi_1(C)) \triangle r(I_2 \cup \pi_2(C)))|], \\
h_i(C) &= \mathrm{E}[|Y_i \setminus (r(I_1 \cup \pi_1(C)) \triangle r(I_2 \cup \pi_2(C)))|].
\end{aligned}
$$

Finally, we set $T = \{\langle d, d \rangle \mid d \in D\}$.

First note that $f = g_i + h_i$ since $X_i \cap Y_i = \emptyset$. The proof of Lemma 2 shows that $g_i$ is monotonically increasing and submodular.

Let $C \subseteq C_{i-1} \cup T$. If there is a vertex $v$ in $r(I_1 \cup \pi_1(C))$ but not in $X_i$, then this means $v$ was influenced by $d \in D$. Since $d \in \pi_2(C)$, we have $v \in r(I_2 \cup \pi_2(C))$. That is,

$$r(I_1 \cup \pi_1(C)) \setminus X_i = r(I_2 \cup \pi_2(C)) \setminus X_i.$$

Since $Y_i$ and $X_i$ are disjoint, this gives us

$$
\begin{aligned}
h_i(C) &= \mathrm{E}\left[|Y_i \setminus (r(I_1 \cup \pi_1(C)) \triangle r(I_2 \cup \pi_2(C)))|\right] \\
&= \mathrm{E}\left[|Y_i \setminus ((r(I_1 \cup \pi_1(C)) \setminus X_i) \triangle (r(I_2 \cup \pi_2(C)) \setminus X_i))|\right] \\
&= \mathrm{E}\left[|Y_i|\right].
\end{aligned}
$$

That is, $h_i(C)$ is constant for any $C \subseteq C_{i-1} \cup T$. Thus, $h_i(C) = h_i(C_{i-1})$.

Finally, the assumption of the proposition guarantees that $f(C_i) \geq f(C_{i-1} \cup \{t\})$, for $t \in T$.

Thus, these definitions meet all the prerequisites of Lemma 6, guaranteeing that

$$(1 - 1/e)\Phi_C(D, D) \leq \Phi_C(S_1^{k/2}, S_2^{k/2}) \leq \Phi_C(S_1^k, S_2^k).$$

Since $OPT/2 \leq \Phi_C(D, D)$, the result follows. $\qquad\square$

# F Additional tables and figures related to the experimental evaluation

Table 1: Dataset descriptions, as well as tags and rewteets that were used to collect the data.

**USelections**: Tweets containing hashtags and keywords identifying the USElections, such as #uselections, #trump2016, #hillary2016, etc. Collected using Twitter 1% sample for 2 weeks in September 2016

| *Pro-Hillary* | *Pro-Trump* |
|---|---|
| RT @hillaryclinton, #hillary2016, #clintonkaine2016, #imwithher | RT @realdonaldtrump, #makeamericagreatagain, #trumppence16, #trump2016 |

**Brexit**: Tweets containing hashtags #brexit, #voteremain, #voteleave, #eureferendum for all of June 2016, from the 1% Twitter sample.

| *Pro-Remain* | *Pro-Leave* |
|---|---|
| #voteremain, #strongerin, #remain, #remaineu, #votein | #voteleave, #strongerout, #leaveeu, #takecontrol, #leave, #voteout |

**Abortion**: Tweets containing hashtags #abortion, #prolife, #prochoice, #anti-abortion, #pro-abortion, #plannedparenthood from Oct 2011 to Aug 2016.

| *Pro-Choice* | *Pro-Life* |
|---|---|
| RT @thinkprogress, RT @komenforthecure, RT @mentalabortions, #waronwomen, #nbprochoice, #prochoice, #standwithpp, #reprorights | RT @stevenertelt, RT @lifenewshq, #praytoendabortion, #prolifeyouth, #prolife, #defundplannedparenthood, #defundpp, #unbornlivesmatter |

**Obamacare**: Tweets containing hashtags #obamacare, and #aca from Oct 2011 to Aug 2016.

| *Pro-Obamacare* | *Anti-Obamacare* |
|---|---|
| RT @barackobama, RT @lolgop, RT @charlespgarcia, RT @defendobamacare, RT @thinkprogress, #obamacares, #enoughalready, #uniteblue | RT @sentedcruz, RT @realdonaldtrump, RT @mittromney, RT @breitbartnews, RT @tedcruz, #defundobamacare, #makedclisten, #fullrepeal, #dontfundit |

**Fracking**: Tweets containing hashtags and keywords #fracking, 'hydraulic fracturing', 'shale', 'horizontal drilling', from Oct 2011 to Aug 2016.

| *Pro-Fracking* | *Anti-Fracking* |
|---|---|
| RT @shalemarkets, RT @energyindepth, RT @shalefacts, #fracknation, #frackingez, #oilandgas, #greatgasgala, #shalegas | RT @greenpeaceuk, RT @greenpeace, RT @ecowatch, #environment, #banfracking, #keepitintheground, #dontfrack, #globalfrackdown, #stopthefrackattack |

**iPhone vs. Samsung**: Tweets containing hashtags #iphone, and #samsung from April (release of Samsung Galaxy S7), and September 2015 (release of iPhone 7).

| *Pro-iPhone* | *Pro-Samsung* |
|---|---|
| #iphone | #samsung |

Table 2: Dataset statistics. The column $|C|$ refers to the average number of edges in a randomly generated cascade in the correlated case, while $|C_1|$ and $|C_2|$ refer to average number of edges generated in a cascade of the campaigns 1 and 2, respectively, in the heterogeneous case.

| Dataset | # Nodes | # Edges | $|C|$ | $|C_1|$ | $|C_2|$ |
|---|---|---|---|---|---|
| Abortion | 279 505 | 671 144 | 2 105 | 326 | 1 801 |
| Brexit | 22 745 | 48 830 | 476 | 113 | 390 |
| Fracking | 374 403 | 1 377 085 | 4 156 | 1 595 | 3 103 |
| iPhone | 36 742 | 49 248 | 4 776 | 339 | 4 478 |
| ObamaCare | 334 617 | 1 511 670 | 6 614 | 2 404 | 4 527 |
| US-elections | 80 544 | 921 368 | 4 697 | 3 097 | 12 044 |

Table 3: Detailed values of the data presented in Figure 2. The data correspond to the absolute value expected symmetric difference $n - \Phi$ of Hedge and the baselines for $k = 20$ across all datasets. Low values are better.

| | Heterogeneous setting | | | | | |
|---|---|---|---|---|---|---|
| Dataset | Hedge | BBLO | Inters. | Union | HighDeg. | Random |
| Abortion | 1436.090 | 1447.710 | 1571.180 | 1655.580 | 3414.310 | 4253.220 |
| Brexit | 17.907 | 17.765 | 31.850 | 27.770 | 54.131 | 87.341 |
| Fracking | 3411.810 | 3420.700 | 3651.230 | 3825.360 | 5197.060 | 7449.350 |
| iPhone | 421.411 | 865.126 | 839.119 | 1048.090 | 1189.650 | 631.543 |
| ObamaCare | 1768.560 | 1828.900 | 1998.250 | 1846.750 | 3315.570 | 4032.140 |
| US-elections | 515.347 | 516.587 | 1030.640 | 685.089 | 1474.330 | 5988.160 |

| | Homogeneous setting | | | | | |
|---|---|---|---|---|---|---|
| Dataset | Hedge | BBLO | Inters. | Union | HighDeg. | Random |
| Abortion | 144.898 | 185.569 | 446.462 | 444.766 | 2368.610 | 1279.100 |
| Brexit | 1.232 | 1.615 | 9.643 | 9.374 | 28.850 | 34.283 |
| Fracking | 275.143 | 269.404 | 1423.870 | 781.994 | 2529.570 | 2960.720 |
| iPhone | 14.624 | 19.893 | 79.854 | 80.279 | 895.353 | 759.629 |
| ObamaCare | 97.319 | 95.062 | 1314.830 | 360.103 | 2253.050 | 2484.330 |
| US-elections | 64.870 | 103.318 | 128.586 | 104.911 | 1979.79 | 5325.130 |

Figure 1: Expected symmetric difference $n - \Phi_H$ as a function of the budget $k$. Heterogeneous model. Low values are better.

Figure 2: Expected symmetric difference $n - \Phi_C$ as a function of the budget $k$. Correlated model. Low values are better.

Figure 3: Running time as a function of number of edges. Correlated model with $k = 20$.

Side 1  Side 2  Hedge
*Pro-Choice*  *Pro-Life*

*Pro-Remain*  *Pro-Leave*

*Pro-Fracking*  *Anti-Fracking*

*Pro-iPhone*  *Pro-Samsung*

*Pro-Obamacare*  *Anti-Obamacare*

*Pro-Hillary*  *Pro-Trump*

Figure 4: Word clouds of the profiles for the initial seeds, and profiles selected by Hedge.