[Reviews · NeurIPS 2017]

Reviewer 1



The paper considers the problem of balancing the information exposure in the social network. The authors consider two opposing views spreading through the network, and pose the problem of selecting nodes for infecting so that the number of imbalanced (i.e. infected by only one of the two views) nodes is minimized. The problem seems important, especially in the prevalent spreading of the fake news. The model assumed by the authors is the independent-cascade model (ICM). It facilitates simple analysis, and it is Ok to start with simple models. However, I would say it is not a very realistic model, and more complex models exist which proved to capture social network phenomena well (e.g. Hawkes processes). I am not particularly convinced by the settings taken by the authors. Why would it be reasonable to assume that we have probabilities for each node reposting a post? Even if we assume different probabilities for the two views, we are assuming independence of the two spreads. Shouldn't we rather have at least four probabilities for each edge? (i.e., probability of reposting story 1 if nothing was reposted yet, probability of reposting story 2 if nothing was reposted yet, probability of reposting story 1 if story 2 was already reposted, probability of reposting story 2 if story 1 was already reposted). That would sound more realistic to me, because if I already reposted supporting view on X, perhaps I am less likely to repost a support for Y? The experimental evaluation is difficult to conduct for such a problem. I wonder how it could be improved to make it look less synthetic (since the real data provided the parameters for the random simulations). Perhaps a bit elaboration on this point would be good for appreciation of the experiments conducted.

Reviewer 2



In this work the authors consider the problem of choosing seed sets subject to cardinality constrainsts such that the number of nodes that will be exposed to both or none of two campaigns will be maximized. The authors consider two different models of independent cascades, the heterogeneous and the correlated model. The authors propose a correlated independent cascade model where p1(e)=p2(e). Would it make sense for competing campaigns to consider a model where p1(e)=1-p2(e)? For instance, one may expect that a fully commited Democrat is less likely to retweet campaign tweets of Republicans, and vice versa. Then, the authors prove that the problem is NP-hard. The reduction is provided in the appendix. The authors provide 3 algorithms, that come with guarantees under certain conditions. The first algorithm decomposes the objective into two terms one of which is submodular, and optimizes this term. It would be nice to have a sentence (say in page 5, starting from line 190) describing in words the semantics of sets \Omega(S_1,S_2), \Psi(S_1,S_2). Since \Omega is submodular, the authors suggest in algorithm 0 to optimize only Omega. The other two algorithms are intuitive but work for the correlated model. Among the 3 proposed algorithms, the last one (Hedge) performs better in practice, even for the setting that the theoretical guarantees do not apply. However the improvement over Borodin's et al method is relatively small. The authors should provide the actual numbers in text since it's hard to tell precisely from fig.2. The experiments are nicely done, including the preprocessing of setting the probabilities on the edges. Can the authors elaborate on when one expects the baseline HighDegrees to perform worse than Random? Also, how different are the cloud of words produced by Borodin et al compared to your method? Some clouds in the Appendix would be helpful to understand not only the performance based on metrics. Minor: (a) Please provide a reference for the #P hardness (e.g., "Scalable Influence Maximization for Prevalent ViralMarketing in Large-Scale Social Networks" by Chen et al.). (b) Make sure to remove all typos (e.g., ctrl+F in appendix rewteets instead of retweets) Overall, this is a solid paper, with interesting experimental findings.

Reviewer 3



This paper proposes the problem of exposure balancing in social networks where the independent cascade is used as the diffusion model. The usage of symmetric difference is quite interesting and the approximation scheme to shovel the problem is plausible. The experiments setup and the way the two cascades are defined and curated are interesting too. It’s not clear if the two social network campaigns should be opposing or not? Is it a necessity for the model? Furthermore, some hashtags are not really in opposition to each other. For example, on obamacare issue, the tag @sentedcurze could represent many many topics related to Senator Ted Cruz including the anti-obamacare. Any elaboration on the choice of \alpha=0 for the correlated setting. Given the small cascades size, is the diffusion network learned reliable enough? It looks more like a heuristic algorithm for learning the diffusion network which might be of high variance and less robust given the small cascades size. In Figure 3 why for some cases the difference between methods is increasing and for some cases it is decreasing? Using the term “correlated” at the first though reminds the case that the two cascades are competing against each other to attract the audience. This looks to be a better definition of correlation than the one used in the paper. But in this paper, it’s not clear until the user reaches to page 3. I would suggest to clarify this earlier in the paper. Showing standard deviations on the bar charts of figure 1 would have helped assess the significance of the difference visually. Furthermore, reporting significance levels and p-values could make the results stronger. A few very related work are missing. Especially paper [1] has formulated the problem of exposure shaping in social networks. Furthermore, paper [2] considers two cascades who are competing against each other and optimizes one of them so they can hit the same number of people, i.e., trying to balance their exposure. Given papers [1] and [2] I would say the current paper is not the first study of balancing g exposures in social networks. They are closely related. Paper [3] has proposed a point-process based model for competing cascades in social networks that is related and finally paper [4] proposes an algorithm to optimize the activities in social networks so that the network reaches to a specific goal. [1] Farajtabar, Mehrdad, et al. "Multistage campaigning in social networks." Advances in Neural Information Processing Systems. 2016. [2] Farajtabar, Mehrdad, et al. "Fake News Mitigation via Point Process Based Intervention." arXiv preprint arXiv:1703.07823 (2017). [3] Zarezade, Ali, et al. "Correlated Cascades: Compete or Cooperate." AAAI. 2017. [4] Zarezade, Ali, et al. "Cheshire: An Online Algorithm for Activity Maximization in Social Networks." arXiv preprint arXiv:1703.02059 (2017).